# Engineering Modified mRNA-Based Vaccine against Dengue Virus Using Computational and Reverse Vaccinology Approaches

**DOI:** 10.3390/ijms232213911

**Published:** 2022-11-11

**Authors:** Mamuna Mukhtar, Amtul Wadood Wajeeha, Najam us Sahar Sadaf Zaidi, Naseeha Bibi

**Affiliations:** Atta-ur-Rahman School of Applied Biosciences, National University of Sciences and Technology, Sector H-12, Islamabad 44000, Pakistan

**Keywords:** dengue virus, mRNA vaccines, reverse vaccinology, molecular docking, normal mode analysis, immune simulations

## Abstract

Dengue virus belonging to the family *Flaviviridae* and its four serotypes are responsible for dengue infections, which extend over 60 countries in tropical and subtropical areas of the world including Pakistan. During the ongoing dengue outbreak in Pakistan (2022), over 30,000 cases have been reported, and over 70 lives have been lost. The only commercialized vaccine against DENV, Dengvaxia, cannot be administered as a prophylactic measure to cure this infection due to various complications. Using machine learning and reverse vaccinology approaches, this study was designed to develop a tetravalent modified nucleotide mRNA vaccine using NS1, prM, and EIII sequences of dengue virus from Pakistani isolates. Based on high antigenicity, non-allergenicity, and toxicity profiling, B-cell epitope, cytotoxic T lymphocyte (CTL), and helper T lymphocyte (HTL) putative vaccine targets were predicted. Molecular docking confirmed favorable interactions between T-cell epitopes and their respective HLA alleles, while normal mode analysis validated high-affinity interactions of vaccine proteins with immune receptors. In silico immune simulations confirmed adequate immune responses to eliminate the antigen and generate memory. Codon optimization, physicochemical features, nucleotide modifications, and suitable vector availability further ensured better antigen expression and adaptive immune responses. We predict that this vaccine construct may prove to be a good vaccinal candidate against dengue virus in vitro as well.

## 1. Introduction

Dengue fever is a viral disease caused by dengue virus, which is transmitted via female mosquitoes, mainly *Aedes aegypti* and *Aedes albopictus* of the family *Flaviviridae*, which includes several other viruses, i.e., yellow fever virus (YFV), West Nile virus (WNV), Japanese encephalitis virus (JEV), and tick-borne encephalitis virus (TBEV) [1,2]. Clinical symptoms and signs among dengue patients include fever, malaise, headache, and asthenia. However, in severe cases, the disease progresses to dengue hemorrhagic fever (DHF), which is characterized by hemostasis abnormalities and increased vascular permeability, the latter potentially leading to hypovolemic shock (dengue shock syndrome, DSS) [3,4,5]. Dengue fever is a serious public health risk for nearly one-third of the world’s population, who live in tropical and subtropical climates. Dengue fever infection has increased dramatically during the last several decades. Based on a World Health Organization (WHO) estimate, 390 million new outbreaks of dengue fever are reported worldwide every year. Dengue incidence in India has surged fivefold in the previous five years, according to the National Vector Borne Disease Control Program (NVBDCP). Dengue fever hits over 100 nations in Asia, the Caribbean, the Americas, the Pacific, and Africa, with 100–400 million infections every year [6,7].

Dengue is endemic in Pakistan, and dengue fever first emerged as the most significant mosquito-borne viral disease in 1994 [8]. From 2006 to 2011, Pakistan faced the worst condition regarding dengue in the post-monsoon period, with 40,987 confirmed cases and 490 deaths [9,10]. In August 2013, a large dengue outbreak occurred in Khyber Pakhtunkhwa Province, affecting more than 7000 people with 26 deaths [11]. In Rawalpindi District, there were 1100, 1406, and 3900 confirmed cases of dengue fever in 2013, 2014, and 2015, respectively, including seven deaths in 2015 [12]. From January 2014 to May 2020, 99,264 confirmed cases were reported from all four major provinces of the country, while in 2019, approximately 25,000 cases were reported [13]. In the ongoing dengue outbreak of 2022, almost 31,000 cases have been reported with a death toll of ~70. 

Dengue virus, like other *flaviviruses*, has an enveloped, ~11 kb genome (positive-sense RNA), with a single long open reading frame (ORF) that encodes for three structural proteins, i.e., (Capsid protein C, precursor membrane protein (prM), and envelope protein E), and seven non-structural proteins, i.e., NS1, NS2B, NS2A, NS4A, NS3, NS4B, and NS5. The genome of dengue virus evolves rapidly, and positive mutations can change the virus phenotypically, resulting in the evolution of severe diseases [14,15]. Dengue infections are generally believed to be caused by four distinct serotypes: DENV-1, DENV-2, DENV-3, and DENV-4, each generating a unique host immune response to the infection. These four serotypes are genetically similar to each other and share approximately 65% of their genomes [16,17]. 

There are several vaccine candidates that are in early clinical and preclinical studies. Five types of dengue vaccine candidates: live attenuated viruses, inactivated whole viruses, viral vectors, subunit vaccines, and DNA vaccines have been reported, and currently, CYD-TDV (Dengvaxia^®^) is the only licensed vaccine [18,19]. WHO recommended the use of this vaccine in individuals aged between 9 and 45 years old [20]. In Asia and Latin America, two additional vaccine candidates, TAK-003 (previously known as DENVax) and TV003/TV005, are being tested in efficacy trials, and TAK-003 has an overall protection rate of 73.3%. The safety concerns of dengue vaccination programs lead to the urgent need for safer and more effective second-generation dengue vaccines [7,21]. 

mRNA vaccines have gained much interest in vaccinology as a promising alternative to conventional vaccine approaches. Two major types include non-replicating mRNA virally derived and self-amplifying RNA. This versatile technology with intrinsic self-adjuvant properties can achieve strong humoral and cellular immune responses. Multiple mRNA vaccines against infectious diseases and several types of cancer are under investigation [22,23]. In this study, three proteins were analyzed, i.e., NS1, prM, and domain 3 of envelope protein (EIII), of four serotypes (DENV 1–4) of dengue virus from Pakistani isolates for developing an mRNA vaccine using computational approaches. To increase translation efficiency, codon optimization of the conserved sequence was performed. T-cell and B-cell epitopes were discovered from the consensus protein, and 3D structures were generated. This research provides the groundwork for future research into producing effective vaccines to prevent dengue infection.

## 2. Results

### 2.1. Identification of Consensus Sequence

All the available sequences of NS1, prM, and E proteins from all four serotypes of dengue virus from Pakistan were retrieved from NCBI and aligned. Multiple sequence alignment of each protein was visualized using web-based software NX4, which calculates the frequency of each nucleotide along the alignment based on the Shannon entropy and shows the areas of high and low divergence (Appendix A). UGENE software generated a consensus sequence for each protein. Aligned sequences were also subjected in Jalview alignment viewer, and similar results as UGENE were obtained. Consensus sequences were reverse-translated for codon optimization.

### 2.2. Codon-Optimized mRNA

The GC content (%) and codon adaptation index (CAI) of optimized mRNAs were obtained from JCat server (Table 1). The free energy of the thermodynamic ensemble (kcal/mol), the lowest free energy of the centroid secondary structure (kcal/mol), and the minimum free energy of the optimum secondary structure (MFE) of codons were calculated using the RNAfold server. The CAI value of optimized codons of NS1, prM, and EIII mRNAs improved from 0.1 to 0.95 (Figure 1). The CAI parameter indicates the value 1 as best, above 0.8 as good, and above 0.6 as an average expressive state of codons. The GC content of the unoptimized nucleotide sequence of NS1 was 42.4%, which increased up to 65% after optimization. Similarly, the GC content in unoptimized nucleotides of prM and EIII mRNAs was 43.6% and 40.2%, which improved to 66.2% and 62.8%, respectively.

These optimized codons were further subjected to RNAfold and Mfold servers for estimation of minimum free energy and prediction of secondary structures. For NS1, the free energy of the thermodynamic ensemble was calculated to be −416.07 kcal/mol, which was less than the energy calculated for the original ensemble, i.e., −231.29 kcal/mol. The MFE of the optimal secondary structure and that of the centroid secondary structure were calculated to be −402.40 kcal/mol and −326.03 kcal/mol, respectively. Mfold also showed the same results as the ΔG, which for the best calculated structure of NS1 was −418.20 kcal/mol (Appendix A). After the optimization, free energies of the thermodynamic ensemble of prM and EIII were recorded to be −182.94 kcal/mol and −112.14 kcal/mol, respectively. Parallel to NS1, greater free energy of the thermodynamic ensemble was recorded in original nucleotide sequences of prM (i.e., −113.22 kcal/mol) and EIII (i.e., −56.37 kcal/mol). The minimum free energies of the optimal secondary structure and centroid structure of prM were recorded to be −177.70 (kcal/mol) and −173.70 (kcal/mol), respectively. The minimum free energies of the optimal secondary structure and centroid structure of EIII were recorded to be −107.80 kcal/mol and −99.50 kcal/mol, respectively. Mfold predicted the ΔG for the best calculated structure of prM to be −177.0 kcal/mol and −104.4 kcal/mol for EIII. The secondary structures of optimized NS1 mRNA, prM mRNA, and EIII mRNA are shown in Figure 2.

### 2.3. Structure Prediction and Validation of Consensus Proteins

Three-dimensional structures of each protein were obtained from trRosetta, which generated five models. The model with highest TM score was picked to be checked for its quality in SAVES v6.0 server. Model 1 For NS1 and EIII and model 2 for prM were selected for structural refinement using GalaxyRefine. Refined models were again subjected to the SAVES server, and their ERRAT scores, Ramachandran plots (Appendix A), and G factors were studied. For NS1, the ERRAT overall quality factor was calculated to be 86.79. The PROCHECK analysis of the NS1 model showed to have 90.6% of residues in the favored region, 8.4% in the allowed region, and 0.3% in the disallowed region. The overall average G factor was 0.09. The overall quality factor and G factor for the prM protein were recorded to be 100 and 0.23, respectively. The Ramachandran plot of the accepted model showed to have 95.7% of residues in the favored region, 3.6% in the allowed region, and 0.7% in the outlier region. For the EIII protein, the Ramachandran plot predicted that 93.8% of residues lie in the favored region, 4.9% in the allowed region, and 1.2% in the outlier region, and the overall G factor was 0.10. These outcomes validated the reliability of the 3D structure prediction. UCSF chimaera was used to view the protein’s three-dimensional structure (Figure 3).

### 2.4. Identification of Linear B-Cell Epitopes

Various linear B-cell epitopes were predicted by the ElliPro tool of the IEDB server. This server provides several prediction methods and graphically represents the position of B-cell epitopes for consensus protein sequence (Appendix A). The BepiPred epitope prediction method predicts the location of linear B-cell epitopes by combining a hidden Markov model and a propensity scale method. The residues above the default threshold value (i.e., 0.35) are predicted to be part of an epitope and colored in yellow on the graph. Chou–Fasman beta-turn prediction predicts the accessibility of beta-turns, and Emini surface accessibility prediction was used, as B-cell epitopes with scores greater than 1.0 indicate an increased probability for being found on the surface, and hence, increased efficacy. To analyze the flexibility and antigenicity of epitopes from the consensus protein, Karplus and Schulz flexibility prediction and Kolaskar and Tongaonkar prediction methods were performed. To identify the presence of hydrophilic regions, Parker hydrophilicity prediction method was used. ABC-pred predicted 16-mer epitopes, and after the assessment of antigenicity, allergenicity, and the immunogenicity profile, two epitopes for NS1, four for prM, and two for EIII proteins were selected (Table 2). Visual illustrations of locations of epitopes on 3D structures of candidate vaccine proteins are provided in Figure 4.

### 2.5. Identification of Conserved T-Cell Epitopes

The NS1, prM, and EIII proteins’ consensus sequences were examined to have conserved T-cell epitopes to evoke an antigenic response. Using the IEDB MHC-I allele-binding prediction tool, a comprehensive HLA allele reference set was used to identify the MHC class-I epitope from the consensus sequence. Based on analyses of their conservation, allergenicity, and antigenicity, epitopes were narrowed down. This resulted in the selection of three epitopes each for NS1 (ETAECPNTNR, VTRLENLMWK, and VVSWKKKELK) and prM (MGLETRTETW, FTIMAAILAY, and DCWCNATSTW) proteins and two epitopes (EIAETQHGTI and KEIAETQHGT) for the EIII protein (Table 3).

To identify MHC-II epitopes, the consensus protein sequences were once again subjected to the IEDB tool, and several epitopes with lower binding scores were selected. These epitopes were analyzed for their antigenicity, allergenicity, and toxicity, and the epitopes with high antigenicity, non-allergenicity, and non-toxicity were shortlisted. Selected epitopes were also checked for having IFN-γ-, IL-4-, and IL-10-inducing ability. Results from all these analyses showed one epitope each for NS1, prM, and EIII (VFTTNIWLKLKERQD, RHPGFTIMAAILAYT, and KIPFEIMDLEKRHVL, respectively) that met all requirements (Table 4). Selected epitopes showed 100% conservancy.

### 2.6. Three-Dimensional Structure of the Selected Epitopes and Their Corresponding Alleles 

The 3D shapes of the chosen T-cell epitopes were created by the trRosetta peptide structure prediction server. This server generated different models for each epitope, which were selected based on high TM scores and were subjected to the SAVES server (Appendix A). The PROCHECK-generated Ramachandran plot demonstrated that the allele models were satisfactory, with 100% of the residues in the favored region and none in the outlier region. The RCSB Protein Data Bank (PDB) server was used to extract the HLA alleles’ crystal structure. These results assured the precision of the predicted structure. Each epitope had a distinct corresponding allele except for two epitopes of prM, which shared a common allele (HLA-B*58:01). The crystallographic structures of MHC-I and MHC-II alleles were saved at RCSB PDB. 

### 2.7. Molecular Docking 

In this study, two docking simulations were performed, one between the consensus protein and dengue receptors and the other among the selected epitopes and their corresponding MHC alleles. The docking results obtained from GalaxyPepdock based on structure similarity scores (TM) around 0.96 for protein–peptide interactions, an interaction similarity value over 50, and an expected accuracy score over 0.70 are presented in Table 5.

Docking between selected HTL epitopes with their respective MHC-II alleles was performed by Cluspro 2.0, and the best models with the lowest binding (weighted score) energy were chosen (Table 6). 

Epitope docking was visualized by Discovery Studio Visualizer. Utilizing discovery studio, the outcomes of docking were examined to determine how precisely the interaction actually occurred. The epitopes of the NS1 protein formed stable interactions with their corresponding MHC-I alleles. The epitope ETAECPNTNR bound with HLA-A*68:01 with six favorable interactions, epitope VTRLENLMWK showed nine favorable interactions with HLA-B*57:01, and epitope VVSWKKKELK showed seven favorable interactions with HLA-A*03:01. The interaction of VTRLENLMWK with HLA-B*57:01 is shown in Figure 5a. The epitopes of the prM protein also showed stable interactions with their corresponding MHC-I alleles. Epitopes DCWCNATSTW and FTIMAAILAY were able to make eight favorable interactions with their corresponding alleles HLA-A*24:02 and HLA-B*53:01, respectively (Appendix A). The interaction between DCWCNATSTW and HLA-A*24:02 is shown in Figure 5b. Epitope MGLETRTETW of prM made seven favorable interactions with HLA-B*58:01. For the EIII protein, epitope EIAETQHGTI formed five favorable interactions with HLA-A*02:03. Seven favorable interactions were observed between KEIAETQHGT and HLA-B*44:02 (Figure 5c). The MHC-II allele HLA-DRB4*01:01 was shared by two epitopes (RSLRPQPTELKYSWK and FQRVLIFILLTAVAP) of NS1 and prM proteins, respectively. The epitope (KIPFEIMDLEKRHVL) of the EIII protein made seven favorable interactions with HLA-DRB1*01:01. A graphical representation of HTL epitopes with their relevant MHC-II alleles is shown in Figure 5d–f.

Each protein was docked with immune receptors associated with dengue virus, i.e., DC-SIGN, mannose-binding receptor (MR), TLR3, and TLR4, and results were analyzed by PDBsum and visualized by Discovery studio. PDBsum results (Appendix A) showed that there were 23 interface residues, 2 salt bridges, 13 hydrogen bonds, and 200 non-bonded contacts between the NS1 protein and DC-SIGN. The number of interface residues between NS1 protein and MR receptor were 26 and 23, respectively, with 10 hydrogen bonds and 160 non-bonded contacts between them. In total, 50 interface residues were observed between the NS1 protein and TLR3 receptor, with 10 salt bridges, 23 hydrogen bonds, and 284 non-bonded interactions. The interactions between 27 interface residues of the TLR4 receptor and 30 interface residues of the NS1 protein were supported by 14 hydrogen bonds and 158 non-bonded contacts. Discovery studio revealed that NS1 had nine favorable interactions with each receptor.

PDBsum showed that there were 22 interface residues of DC-SIGN interacting with 30 residues of the prM protein, and there were 1 salt bridge, 12 hydrogen bonds, and 194 non-bonded contacts between them. The number of interface residues between the MR receptor and prM protein were 28 and 29, respectively, with 5 salt bridges, 11 hydrogen bonds, and 207 non-bonded contacts. The number of interface residues observed between TLR3 and the prM protein were 30 and 27, respectively, and there were 2 salt bridges, 13 hydrogen bonds, and 242 non-bonded interactions. The interactions between 40 interface residues of the TLR4 receptor and 38 interface residues of the prM protein were supported by 10 salt bridges, 20 hydrogen bonds, and 273 non-bonded interactions. Discovery studio showed that prM also had nine favorable interactions with DC-SIGN, MR, and the TLR4 receptor and eight favorable interactions with TLR3. 

Sixteen interface residues of both DC-SIGN and the EIII protein interacted with each other, and 1 salt bridge, 7 hydrogen bonds, and 120 non-bonded contacts were observed between them. EIII and MR receptors showed 14 and 19 interacting interface residues, respectively, and 2 salt bridges, 10 hydrogen bonds, and 146 non-bonded contacts were present between them. TLR3 had 37 interface residues interacting with 21 interface residues of EIII, and they formed 9 salt bridges, 21 hydrogen bonds, and 217 non-bonded contacts between them. The number of interface residues between TLR4 and EIII were 25 and 18, respectively, and 3 salt bridges, 11 hydrogen bonds, and 125 non-bonded atoms were found between them. Discovery Studio analysis revealed that EIII had eight favorable interactions with each immune receptor.

The results of docking based on the lowest docking score between immune receptors and each of the candidate vaccine proteins (ligand) are given in Table 7, and the visual illustrations obtained via discovery studio are presented in Figure 6.

### 2.8. Normal Mode Analysis (NMA) 

Simulations were performed between candidate dengue virus proteins and immune receptors in normal mode analysis (NMA), and attained results of NS1–MR, prM–TLR3, and EIII–TLR4 docking complexes are depicted in Figure 7, Figure 8 and Figure 9, respectively. Hinge points on the complexes’ deformability graphs serve as indicators of the complexes’ deformability regions and are illustrated in Figure 7b, Figure 8b and Figure 9b. Eigenvalues of the NS1–MR complex was found to be 8.377077 × 10^−5^ and for the prM–TLR3 and EIII–TLR4 docking complexes were recorded to be and 1.627659 × 10^−5^ and 2.392051 × 10^−5^, respectively (Figure 7c, Figure 8c and Figure 9c). The uncertainties of the atoms in the docking complex are represented by the B-factor, which computes the root mean square value (Figure 7d, Figure 8d and Figure 9d). Figure 7e, Figure 8e and Figure 9e show the covariance matrix, which depicts the linkage between pairs of residues that have correlated (red), uncorrelated (white), and anti-correlated (blue) motions. The relationship between the atoms is explained by the elastic docking network (dark gray) (Figure 7f, Figure 8f and Figure 9f). The results of the iMODS simulation point to the stability of immune receptor complexes and vaccine constructions (Appendix A).

### 2.9. Physicochemical Properties of the Conserved Protein

To estimate the physicochemical properties of consensus protein sequences, the Protparam server was explored. Proteins were assessed for parameters such as theoretical pI, extinction coefficient, instability index, aliphatic index molecular weight, amino acid composition, and grand average of hydropathicity (GRAVY). The allergenicity of candidate vaccine proteins was assessed by AllerTOP v.2.0, which predicted that NS1 was a probable allergen, whereas prM and EIII proteins were non-allergenic in nature. Antigenicity of each of the candidate proteins was estimated by Vaxijen v.2.0, with the viral threshold set at 0.4, and it was projected that all of the vaccine proteins would be antigenic. The ToxinPred server predicted that the vaccine proteins were non-toxic. The instability index revealed that NS1 and prM were slightly unstable, while the EIII protein was stable. The pI value of proteins showed that NS1 was slightly acidic, prM was acidic, and EIII was basic in nature. A high aliphatic index indicates the thermostability of proteins, while the hydrophilic character of each protein is shown by a negative GRAVY score (Appendix A).

### 2.10. In Silico Immune Simulations

Immune simulations results for NS1, prM, and EIII are shown in Figure 10, Figure 11 and Figure 12, respectively. According to an immune simulation study, the primary reaction was dominated by the secondary and tertiary responses. Increased IgM levels were the primary response, while IgM + IgG, IgG1 + IgG2, IgG1, IgG2, and B-cell populations were the secondary responses (Figure 10a, Figure 11a and Figure 12a). This demonstrated the emergence of immunological memory and effective immunity with recurrent antigen encounters. Potential for antibody class switching and memory was indicated by the long-term presence of several B-cell isotypes (Figure 10b, Figure 11b and Figure 12b). Similarly, higher responses were noticed in the helper T lymphocytes as shown in Figure 10c, Figure 11c and Figure 12c, and high cytotoxic T lymphocyte populations with development of their respective memory were noticed (Figure 10d, Figure 11d and Figure 12d). Increased macrophage activity and consistent dendritic cell activity were observed (Figure 10e, Figure 11e and Figure 12e). Moreover, elevated levels of interferon gamma (IFN-γ) and interleukin-2 (IL-2) and elevated concentrations of cytokines (IFN-c) and interleukins were also recognized after three injections were administered periodically (Figure 10f, Figure 11f and Figure 12f). Furthermore, innate immune system components (e.g., epithelial cells) were also active, and a reduced Simpson index (D) suggested the likelihood of a range of immunological responses.

### 2.11. Design of Vaccine and Plasmid Construct

The final design of the mRNA vaccine construct for each target sequence is shown in Figure 13a. 5’ Cap 1 (7mGpppN20-Om-RNAs) in the design will cloak mRNA from enzymatic degradation and improve translation, whereas a 120-base poly (A) tail will improve mRNA longevity and translation rate. The whole sequences of the proposed mRNA vaccines, from N-terminus to C-terminus, are documented (Appendix A). The SnapGene tool was utilized to design a plasmid DNA structure with the NS1–mRNA vaccine. The prepared RNA vaccine cDNA (1571 bp) was inserted into the pET28a(+) circular plasmid vector between the XbaI restriction site at the N-terminal and the Hind-III restriction site at the C-terminal (Figure 13b), and the length of total recombinant plasmid was 6784 bp. The cloned plasmid DNA had a T7 bacteriophage promoter, an open reading frame (ORF) encoding target protein flanked by 5’ and 3’ UTR sequences, a poly [d(A/T)] sequence, and a specific restriction site for plasmid linearization to guarantee specified end of transcription. An antibiotic resistance gene and an Ori region were also included in the plasmid.

## 3. Discussion

Currently, Pakistan, along with the rest of the world, is fighting against the COVID-19 crisis and facing the challenges of dengue infection breakouts in many areas. The dual burden of dengue and COVID-19 is having drastic effects on the population as well as the nation’s healthcare system. The country experienced the third wave of COVID-19, which surged in March–April 2021, raising the total number of cases and fatalities to 941,170 by June 2021 [24,25]. Pakistan’s environmental conditions and geographical location are favorable for breeding of the *Aedes aegypti* mosquito vector for dengue, which can propel Pakistan’s dengue crisis to an alarming trend as being observed in India and Brazil [26]. In many Asian countries and in Latin America, catastrophic effects of the COVID-19 outbreak have been reported during the dengue season [27]. According to the World Health Organization’s reports, the number of dengue cases increased more than eightfold over the last two decades. Due to climate changes that promote the replication of host mosquitoes, the spread of DENV infections increased from 505,430 in 2000 to over 2.4 million in 2010 to 5.2 million in 2019 [28,29]. Thus, emergency preparedness is necessitated to avoid drastic impacts on Pakistan’s healthcare system. 

To decrease dengue infections worldwide, the first and only licensed vaccine Dengvaxia was developed by Sanofi Pasteur using the backbone of yellow fever virus. Unfortunately, the vaccine has pathogenic effects on dengue-seronegative patients and cannot be administered in individuals less than age 9 or greater than age 45 [30,31,32]. Each of the four dengue serotypes can cause the full spectrum of dengue fever, and due to pre-existing heterotypic DENV antibodies, there is a chance of more severe disease upon secondary infection. Thus, scientists are extensively studying this virus and trying to develop a potential vaccine that can protect against all four DENV serotypes [33]. Having high potency, rapid and cost-effective manufacturing, and safe administration, mRNA vaccines represent a promising alternative to conventional vaccine approaches. The mRNA coding region can be easily manipulated during vaccine development to combat viral infections, which is a newer approach as compared to traditional vaccine approaches. mRNA vaccines produce large amounts of target proteins in immune cells, which is crucial for efficient immunization [23,34]. 

In this study, by using immunoinformatic approaches, nucleotide-modified mRNAs of NS1, prM, and domain III of E proteins were treated as target antigens for the development of vaccines against dengue fever. NS1 has been known for its role in disease pathogenesis and the generation of a strong humoral immune response after infection. DENV NS1 also serves as a diagnostic marker because it functions as a complement-fixing protein for several complement pathway components; therefore, these interactions have an impact on complement activity and complement-mediated neutralization [35,36]. The prM protein serves as a chaperon of E protein, preventing E protein from premature fusion or inactivation within the secretory pathway of the host T cell. E protein participates in cell recognition and cell entry. It consists of three structural domains, i.e., DI, DII, and DIII. The domain III participates in host T-cell binding and stimulates host immunity responses by inducing protective and neutralizing antibodies [37,38]. To construct an mRNA vaccine against ZIKV, the membrane and envelope protein are excellent targets. Another study showed that the mRNA vaccine against ZIKV generated neutralizing antibody titers 50–100 times higher than other vaccine candidates [39,40]. Therefore, the dengue NS1, prM, and EIII proteins are important antigens for vaccine development and diagnostic purposes. 

Firstly, candidate consensus sequences were reverse-translated and codon-optimized (i.e., optimize the fraction of G and C bases) to enhance their translational efficiency in the host expression system. Codon optimization has applications in recombinant protein therapeutics and nucleic acid treatments such as gene therapy, mRNA therapy, and DNA/RNA vaccinations [41,42]. To find the immunogenic potential of selected vaccine candidates, several B-cell and T-cell epitopes were identified. B-cell epitopes are the antibody binding sites on antigens. T-cell epitopes bind well with MHC class-I molecules, and they have the potential to elicit a primary immunological response in the hosts [43]. B-cell epitopes and dengue-specific CD8+ and CD4+ lymphocytes are not only involved in pathogenesis and immunological research but also are main targets for vaccine and diagnostic reagent development against dengue virus [44,45].

Molecular docking between T-cell epitopes and their respective alleles confirmed that the epitopes were good binders with their alleles, and favorable interactions were observed between them. Molecular docking between proteins and immune receptors revealed that proteins were capable of binding with receptors, i.e., TLR3, TLR4, DC-SIGN, and MR (Appendix A). TLRs play a key role in regulating the inflammatory response against infectious viruses. Experimental studies suggested that the interaction of nucleotide-modified mRNAs with Toll-like receptors resulted in low immunogenicity and higher stability in mice [46]. Increased levels of all TLRs were observed both in dengue cases, confirming the participation of TLRs in the disease pathogenesis [47]. DENV entrance in these cells is mediated by DC-SIGN and MR working cooperatively. DC-SIGN is usually found on the surface of the cell, and its expression in many cell lines makes it receptive towards DENV infection. Together, MR and DC-SIGN collect the virus on the cell surface and deliver it to a receptor, guiding it to the endosomal compartment [48]. These arguments predict that mRNA taken up by the host will be identified by Toll-like receptors (TLRs) and other receptors and move away from the phagosome. The mRNA is then translated into a specific antigen inside the cytoplasm. MD simulations in normal mode analysis were performed between immune receptors and candidate vaccine proteins [49]. Results from the iMODs simulation study suggested that vaccine constructs and immune receptor docking complexes are stable.

The physicochemical properties of the proteins assessed using Protparam revealed that NS1 and prM were slightly unstable and acidic in nature, while the EIII protein was stable and showed a basic nature. A significant aliphatic index and a low GRAVY score implied that each protein is thermostable and hydrophilic (Appendix A). The immune simulation study divulged the development of immunological memory and efficacious immunity and recommended that the primary and secondary immunological responses are congruent with the anticipated response. The cytokine IFN-γ was reported to be associated with B-cell proliferation and Ig isotype swapping [50], and IFN-γ is also engaged in antiviral replication; therefore, it can assist cell-mediated immunity as a primary effector molecule [49]. Similar results were shown in the studies conducted by [51]. Hence, these results are significant for the immune response against DENV infections.

mRNA vaccines have gained limelight in recent years as the *Pfizer* and *Moderna* vaccines against COVID-19 have been authorized by the FDA and similar authorities around the world. Clinical trial data indicated that the Pfizer and Moderna vaccines were 95.0% effective in preventing SARS-CoV-2 infection, but the Johnson & Johnson vaccine was only 66.0% effective [52,53,54]. For the first time, these vaccines were shown to be safe. Because of their completely synthetic nature, low cost, flexible development, long-term protectivity in animal models and early human clinical trials, and no interaction with the genome, mRNA-based vaccines have been approved for use on healthy populations [55,56]. Studies suggest that the localization of mRNAs is important for cell growth, cell fate determination, and optimization of protein expression. Retention of mRNA in the nucleus can regulate or buffer the quantity of translated proteins from each mRNA [57]. Use of mRNA-based immunization has been limited by several factors, but intensive efforts have resulted in the development of stable constructs and efficient and optimized delivery systems, and enhanced immunogenicity codon optimization for improvement of GC content can increase the translation efficiency. The use of modified nucleosides such as pseudo-uridine or N-1-methylpseudouridine to eliminate intracellular signaling cues for protein kinase R (PKR) initiation results in improved antigen expression and adaptive immune responses. To provide efficient protection from degradation by nuclease activity, nature’s fleeting molecule, mRNA, is packed in lipid nanoparticle (LNP) formulations [58]. A preclinical investigation has recently shown that mRNA-based vaccines can produce both preventive and therapeutic anticancer immunity. In mice and pigs, mRNA vaccinations can elicit protection against viral infection [59].

## 4. Materials and Methods

### 4.1. Proteome Retrieval

To construct potential mRNA vaccines, three viral proteins with complete CDS were selected. Sequences of NS1, prM, and E protein of all four serotypes of dengue virus from Pakistani isolates were retrieved from the NCBI database in FASTA format. Graphical illustration of workflow of development of mRNA vaccines against dengue virus is given in Figure 14.

### 4.2. Multiple Sequence Alignment (MSA)

Multiple sequence alignment (MSA) was performed using ClastalW and MAFFT version 7.0. The consensus sequences of each protein were obtained using UGENE software tool. UGENE is a software that helps users in viewing, analyzing, and annotating biological data in the form of NSG assemblies, multiple sequence alignment, phylogenetic trees, etc. [60]. All 34 selected sequences of the NS1 protein, 20 sequences of prM, and 180 sequences of the EIII protein were multiply aligned, and a consensus sequence for each protein was obtained. Multiple sequence alignment was visualized using Jalview and web-tool NX4 [61,62]. 

### 4.3. Codon Optimization and Secondary Structure Prediction

Each protein’s nucleotide sequences were codon-optimized for improved production in the host expression system. For the assessment, several metrics were employed, including GC content, the frequency of optimum codon (FOP) in *Homo sapiens*, the minimal free energy (MFE), and the codon adaptation index (CAI). Codon optimization of the NS1, prM, and EIII nucleotides was accomplished in two different online tools, i.e., Java Codon Adaptation Tool server (JCat) [63] and GenSmart Codon Optimization by GenScript [64]. RNAfold and UNAfold webservers were explored for prediction of secondary structures and minimum free energies of the optimized sequences [65].

### 4.4. Prediction and Visualization of Protein Tertiary Structures

To obtain the tertiary structures of NS1, prM, and EIII, the consensus protein sequences were separately subjected to trRosetta webserver and were further refined using GalaxyWebrefine server [66,67]. For the refinement of tertiary structures, the server optimizes the hydrogen bonding network and reduces energy at the atomic level. To validate the predicted structure, the SAVESv6.0 server was used. The ERRAT score estimated the 3D structure with a higher quality factor [68,69]. Additionally, the server produced a Ramachandran plot to show where the amino acids were located in the allowed and disallowed areas. The stereochemical characteristics were measured by the G factor and log-odds score (torsion angles and covalent geometry). The predicted tertiary structure was viewed by UCSF chimera.

### 4.5. Prediction and Assessment of B-Cell Epitopes

Three B-cell epitope prediction methods were used to identify potential B-cell epitopes from the chosen proteins of four dengue serotypes. ABCpred, BCPRED, and ElliPro of the IEDB server were used for the prediction of B-cell epitopes. For a graphical representation of the positions of B-cell epitope sequences for the dengue virus, Kolaskar and Tongaonkar antigenicity, Emini surface accessibility, Karplus and Schulz flexibility prediction, Chou–Fasman beta-turn, and Parker hydrophilicity prediction tests for the presence of hydrophilic regions [70] were performed. From each protein, linear epitopes of 16 residues were selected for antigenicity, allergenicity, and toxicity assessment.

### 4.6. Prediction and Assessment of CTL and HTL Epitopes

Cytotoxic T lymphocyte (CTL) epitope prediction was performed by the IEBD webserver. Using a complete HLA allele reference set, binding epitopes for MHC-I were predicted. Each of the various anticipated epitopes receives a score from the server, and an increased score denotes a greater binding affinity. Using the MHCpred website, which was configured to have an IC50 below the cutoff value of 250 nm, epitopes were also predicted. The IEDB site was used to anticipate helper T-cell lymphocyte (HTL) epitopes. Using the IEDB-recommended 2.22 consensus technique and all the HLA allele reference sets, MHC-II-binding epitopes were predicted [71]. A low score implies being excellent binders, and the server displays an adjusted rank of the anticipated epitopes. The IFN epitope, IL4pred, and IL10pred servers, respectively, were used to extensively examine the epitopes with a lower adapted score for their inducibility of interferon-γ (IFN-γ), interleukin-4 (IL-4), and IL-10 (IL-10) [50]. These criteria were then applied to the remaining epitopes, and only the antigenic and three cytokine-inducing epitopes were chosen. After the selection of HLA-restricted CD8+ and CD4+ T-cell epitopes, peptides were assessed for their toxicity, allergenicity, and antigenicity profiles from the ToxinPred server, AllergenFP 1.0, and Vaxijen 2.0 [71]. The antigenicity index was generated at a threshold of 0.4. Conservation analysis of selected CD8+ and CD4+ epitopes was performed using IEDB. While the identity refers to the degree of correspondence (similarity) between the sequences, the conservancy describes the portion of a protein sequence that includes the epitope.

### 4.7. Three-Dimensional Structure Design of Epitopes and HLA Molecules

The selected T-cell epitope candidates were subjected to the online prediction server, trRosetta, which generated 5 models for each submitted peptide sequence. The best model was selected based on the highest confidence level with a higher estimated TM score [72]. The RCSB Protein Data Bank (PDB) server yielded the HLA molecules’ crystalline structure where the allele was coupled with an epitope [73]. Therefore, Discovery studio was used to separate the protein and peptide after which the protein molecules were docked with the selected epitopes [74]. 

### 4.8. Molecular Docking 

To determine how some immunogenic proteins interacted with human immune receptors, including Toll-like receptors 3 and 4, DC-SIGN, and mannose-binding receptor (MR), molecular docking was used. An immunogen interacts with an immune receptor molecule and activates a downstream signaling cascade, which eventually triggers the immune responses. The ClusPro 2.0 server was used for the protein–protein docking between receptor and candidate proteins [75]. The crystal structures of receptors were retrieved from RCSB PDB. Pre-processing of complexes, i.e., water and ligand removal from receptor structures, were performed before subjecting them to the server [30]. The server generated ten different models for the docked structures, and their energy scores were also given. The model with lowest energy values was picked as the best.

The GalaxyPepDock server was used for docking simulation between the selected epitopes (ligand) and their respective MHC-I allele (HLA molecule) [76,77], and ClusPro 2.0 was used for docking between epitopes and MHC-II alleles. The GalaxyPepDock server, a database of empirically confirmed structures, searches for a similar protein–peptide interaction and provides ten conformational structures for each epitope based on their dock scores and interaction similarity scores with the corresponding alleles. This pattern is used to improve the docking if similar peptide–protein interactions are discovered. Based on how closely the top 10 docking conformations resemble the template, the server reports them. Unfortunately, no docking scores were published; instead, GalaxyPepDock predicts the critical amino acid residues required for the attachment of the HLA molecule to various epitopes based on the closeness of the interactions between the template and anticipated docked model [78,79]. Docking models with the highest interaction similarity scores were selected and visualized using Discovery Studio. 

### 4.9. Normal Mode Analysis (NMA)

To check the physical movement and stability of atoms in the docking complex, a molecular dynamics simulation study was conducted using the iMODS server. MD simulations were performed between candidate vaccine proteins and human immune receptors. The iMODs online server was chosen to complete the normal mode analysis (NMA) study, the investigation of the structural dynamics of the docking complex, and determination of the molecular motion [80]. The iMODS server is a convenient, customable, easy, and user-friendly online server to use. The web server takes docked PDB as an input file and generates complex deformability, B-factor, eigenvalues, variance, a covariance map, and an elastic network. The results are displayed keeping all the parameters as default [81].

### 4.10. Assessment of Physicochemical Properties of Translated Protein

Various physicochemical properties of the target proteins such as amino acid composition, instability index, aliphatic index, and grand average hydropathicity (GRAVY) were assessed using the Protparam tool [82]. Antigenicity and toxicity were measured using Vaxijen and ToxinPred, respectively.

### 4.11. In Silico Immune Simulation

The C-ImmSim server was used to predict and estimate the real-life immunogenicity profile of the mRNA vaccines [83]. This simulator predicts peptide-binding interactions of the major histocompatibility complex (MHC) and immune responses using a position-specific scoring matrix (PSSM) and machine learning [84]. Most vaccines call for a minimum of 4 weeks to pass between the first and second dosage. Therefore, in this trial, three injections were given that were spaced four weeks apart, each comprising 1000 vaccine construct units. The “time-step” scale is used by the C-ImmSim server to determine how long simulations should last. Each time step on the scale corresponds to eight hours in actual time. The simulation was run with a total of 1050 time steps (for 350 days), with the two injection sites placed at steps 84 and 168, respectively. Some settings were left as defaults.

### 4.12. Designing the Vaccine Construct

To design a highly immunogenic mRNA vaccine, the consensus proteins of NS1, prM, and EIII sequences were reverse-translated, and their codon-optimized mRNAs were used. The construct needed to be designed using an appropriate 5′ cap, 5′-untranslated region (5′-UTR), Kozak sequence (GCC-(A/G)-CCAUG), tissue plasminogen activator (tPA) signal sequence, mRNA sequence, 3′-untranslated region (3′-UTR), and a poly (A) tail at the 3′ end for our mRNA vaccine model. 

#### 4.12.1. Capping

Capping is essential to improve the stability and translational potential of mRNA vaccines. mRNAs from the incomplete viral genomes and eukaryotic mRNAs have a 7-methylguanosine (m7G) cap at its 5′ end (m7GpppN structure) [85]. Translation in most of the eukaryotic mRNAs is initiated with a cap-binding protein, which was identified by Sonenberg et al. and later named eIF4E. Recognition of caps during the early stages of protein translation plays a pivotal role in the regulation of translation [86].

There are three main cap structures depending upon the degree of methylation, i.e., cap 0, 1, and 2. The first nucleotide of cap 1 has a methylated 2′-OH to connect with the 5′ end of the mRNA with a cap. Cap 1 has the power to diminish the stimulation of pattern recognition receptors (PRRs), increasing the effectiveness of mRNA translation [87].

#### 4.12.2. Untranslated Regions (UTRs)

Incorporation of untranslated regions (UTRs) plays a vital role in the decay and translational efficiency of mRNA. Protein translation is facilitated by the presence of regulatory elements in the 5′-UTR and by the length of the 3′-UTR [29]. In an mRNA vaccine, 5′-UTR efficiently loads ribosomes onto the mRNA for translation initiation. Optimization of 5′-UTR and the downstream 3′-UTR enhances mRNA stability. Burrowing of 5′-UTR from the human α-globin gene in Pfizer/BioNTech mRNA vaccines accounts for efficient translation of vaccines. The same approach has been used for designing 3′-UTR by incorporating stability regulatory elements from human α-globin and β-globin genes [88].

#### 4.12.3. Modified Nucleotides

Modified nucleosides in mRNA vaccines against infectious diseases is the latest technique that offers advantages for the generation of potent and long-lived antibody responses. Nucleotide modifications provide additional base-pair stability, giving rise to a high degree of secondary structures, which improves mRNA translation [89]. The altered nucleobases preclude mRNA vaccines from being detected by the immune system and thus restrict any unintended immunological activation. It is also reported that modified nucleotides under certain circumstances enhance protein translation, thus producing more antigens without triggering harmful side effects such as anaphylaxis [90]. N1-methylpseudouridine-5′-triphosphate is currently preferred over uridine triphosphate, and nearly all mRNA vaccines are being created utilizing m1Ψ instead of uridine.

#### 4.12.4. Poly (A) Tail

The poly (A) tail is crucial for stability and improved protein translation of mRNA, and the length of the poly (A) tail is proportional to translation efficiency. It is also a critical factor for long-term survival of mRNA [89]. Polyadenylation of IVT mRNA can either be done by encoding the poly (A) on the DNA template used to obtain a defined length of poly (A) tail or by enzyme poly (A) polymerase after IVT. Enzymatic polyadenylation of the mRNA results in variable poly (A) tail lengths and is therefore less advantageous [91]. Experimentations have shown that the gradual increase in the shorter length of the poly (A) tail of IVT mRNA enhances the protein expression level, while increasing the number of bases does not further modify protein expression; therefore, a length of 120 bases for the poly (A) tail was chosen [92]. 

### 4.13. In Silico Cloning of Vaccine Construct 

Plasmid (pET-28a(+)) was chosen for the in vitro transcribed mRNA construct rather than a linear plasmid [93]. The replication of origin, kanamycin resistance gene, T7 promoter, terminator site, and several cloning sites are all present on the pET-28a(+) plasmid (MCS). The target sequence received restriction sites, XbaI and Hind-III, at the N- and C-termini, respectively. The vaccine construct’s final nucleotide sequence was generated using the SnapGene tool and cloned into the pET-28a(+) circular vector [94].

## 5. Conclusions

From the immunoinformatic analyses, it is concluded that our designed tetravalent modified mRNA-based dengue vaccine can generate effective immunogenicity against dengue virus infections. Large-scale, cost-effective production with a well-defined product characterization can meet the growing demands of mRNA vaccines for various infectious diseases.

## Figures and Tables

**Figure 1 ijms-23-13911-f001:**
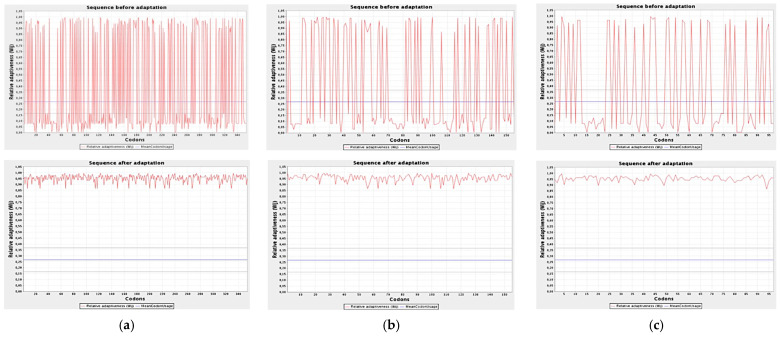
CAI value before optimization (above) and after optimization (below). (**a**) CAI of NS1 improved from 0.1 to 0.95, indicating good expressive state of codons; (**b**) CAI value of prM improved from 0 to 1, indicating the best expressive state of codons; (**c**) CAI value of EIII improved from 0.1 to 0.95, showing good expressive state of codons.

**Figure 2 ijms-23-13911-f002:**
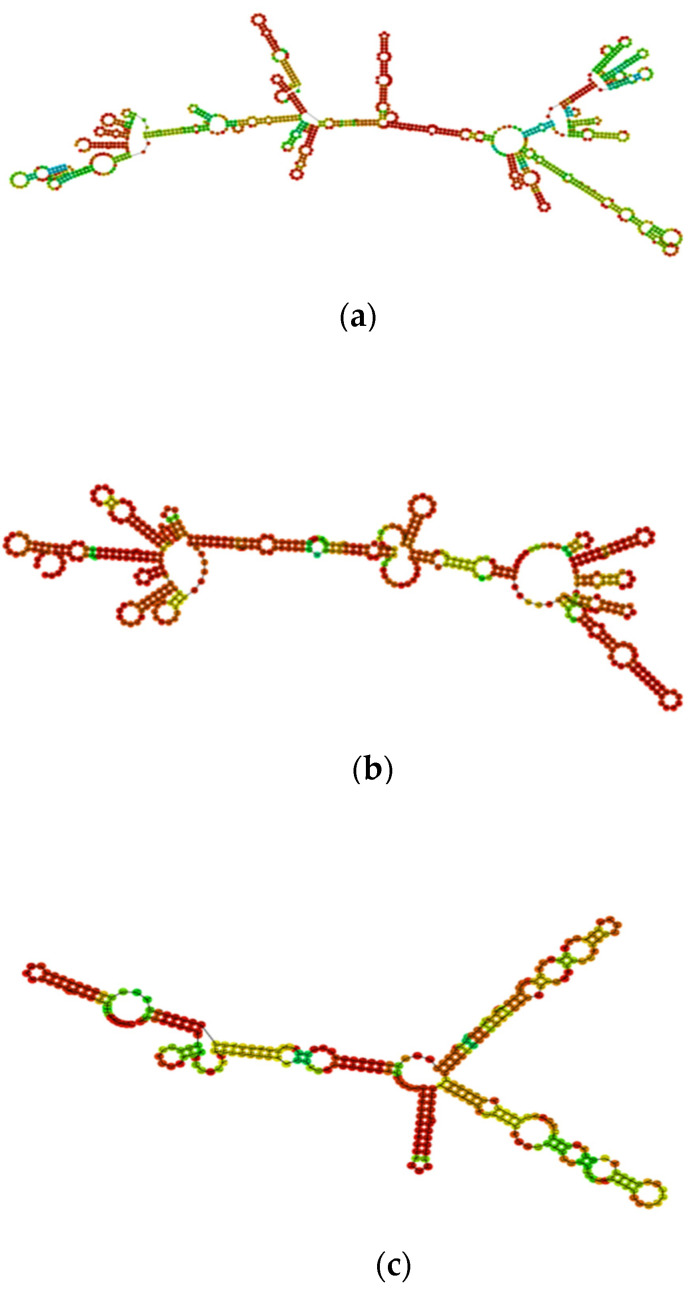
The secondary structures of optimized (**a**) NS1 mRNA; (**b**) prM mRNA; (**c**) EIII mRNA were predicted by the RNAfold server. The structure is colored by base-pairing probabilities. For unpaired regions, the color denotes the probability of being unpaired.

**Figure 3 ijms-23-13911-f003:**
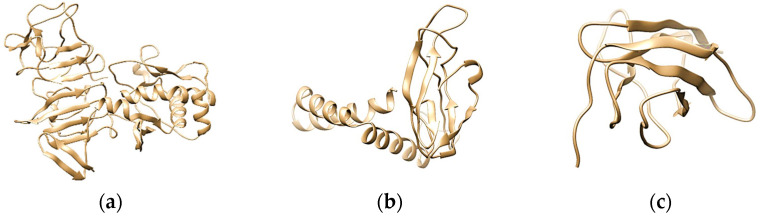
The 3D structure of (**a**) NS1; (**b**) prM; (**c**) EIII consensus proteins predicted by the trRosetta and visualized by UCSF chimera.

**Figure 4 ijms-23-13911-f004:**
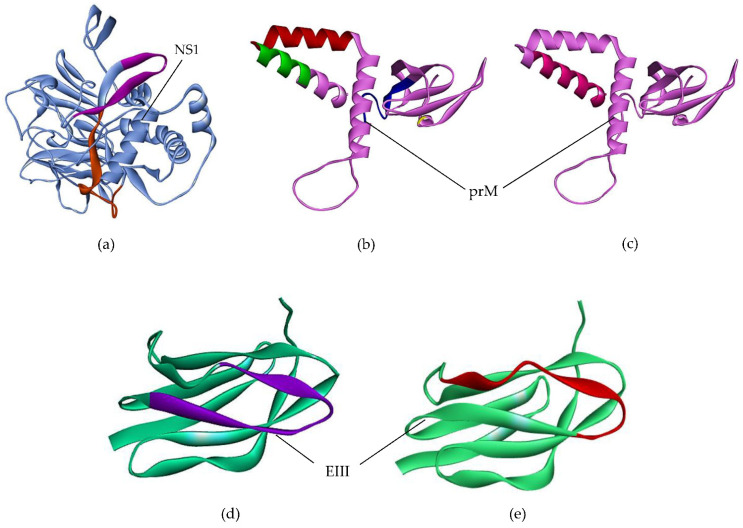
Illustration of Location of epitopes on 3D structures of candidate vaccine proteins. (**a**) Selected B-cell epitopes on NS1 protein, PETAECPNTNRAWNSL (red) and YGFGVFTTNIWLKLKE (purple); (**b**) Selected B-cell epitopes on prM protein, YGTCTATGEHRREKRS (blue), AYTIGTTYFQRVLIFI (red), GFTIMAAILAYTIGTT (green); and (**c**) RVLIFILLTAVAPSMT (pink); (**d**) Selected B-cell epitopes on EIII protein, GRLITVNPIVTEKDSP (purple) and (**e**) NPIVTEKDSPVNIEAE (red).

**Figure 5 ijms-23-13911-f005:**
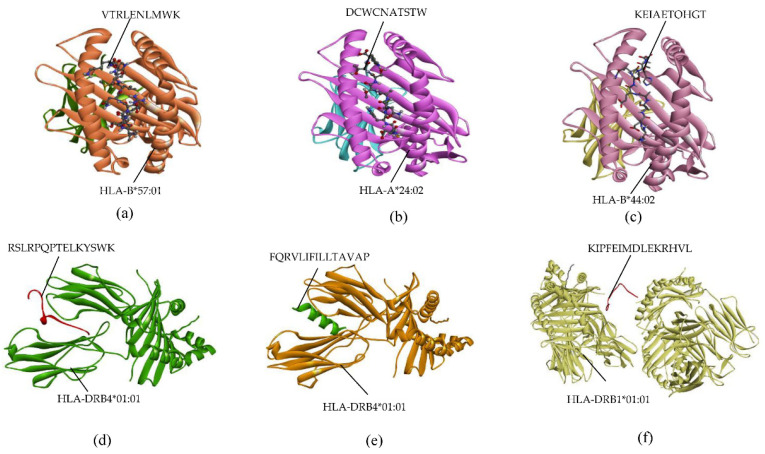
Graphical illustration of docking of epitopes with their corresponding MHC alleles. (**a**) Epitope VTRLENLMWK with HLA-B*57:01. (**b**) Epitope DCWCNATSTW and HLA-A*24:02. (**c**) Epitope KEIAETQHGT and HLA-B*44:02. (**d**) Epitope RSLRPQPTELKYSWK with HLA-DRB4*01:01. (**e**) Epitope FQRVLIFILLTAVAP and HLA-DRB4*01:01. (**f**) Epitope KIPFEIMDLEKRHVL and HLA-DRB1*01:01.

**Figure 6 ijms-23-13911-f006:**
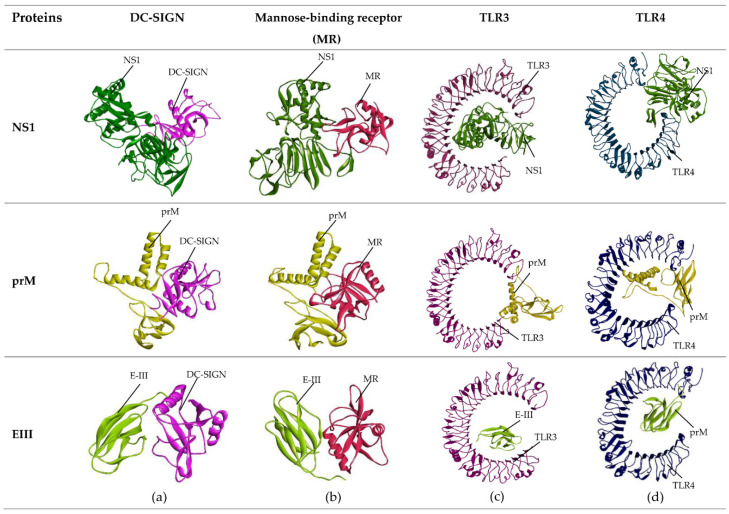
Docking complex of NS1, prM, and EIII proteins and immune receptors. (**a**) DC-SIGN immune receptor complex with candidate vaccine proteins; (**b**) mannose-binding receptor (MR) complex with candidate vaccine proteins; (**c**) TLR3 immune receptor complex with candidate vaccine proteins; (**d**) TLR4 immune receptor complex with candidate vaccine proteins.

**Figure 7 ijms-23-13911-f007:**
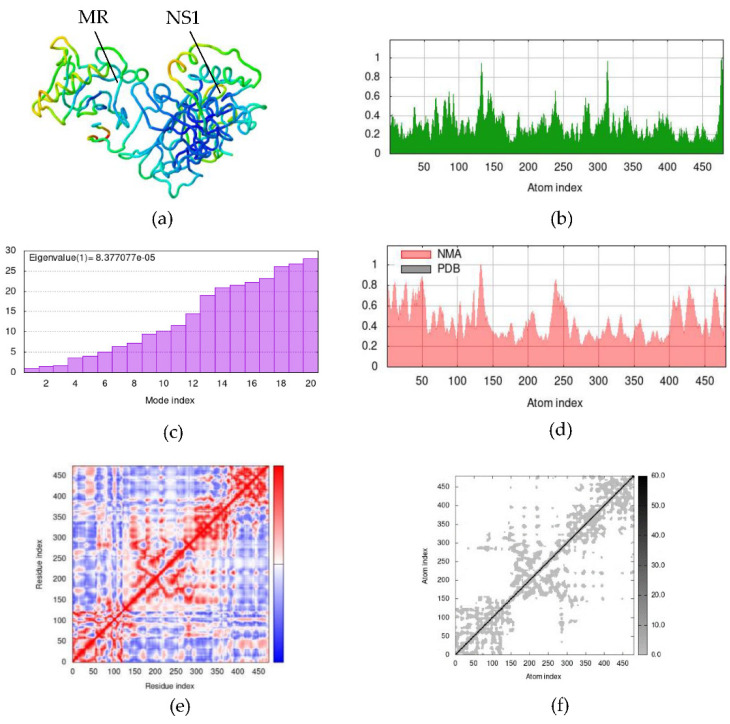
Normal mode analysis (NMA) of NS1–MR docking complex by iMODs. (**a**) NS1 and MR docking complex; (**b**) main-chain deformability; (**c**) the eigenvalue; (**d**) B-factor values; (**e**) covariance map; (**f**) elastic network of model.

**Figure 8 ijms-23-13911-f008:**
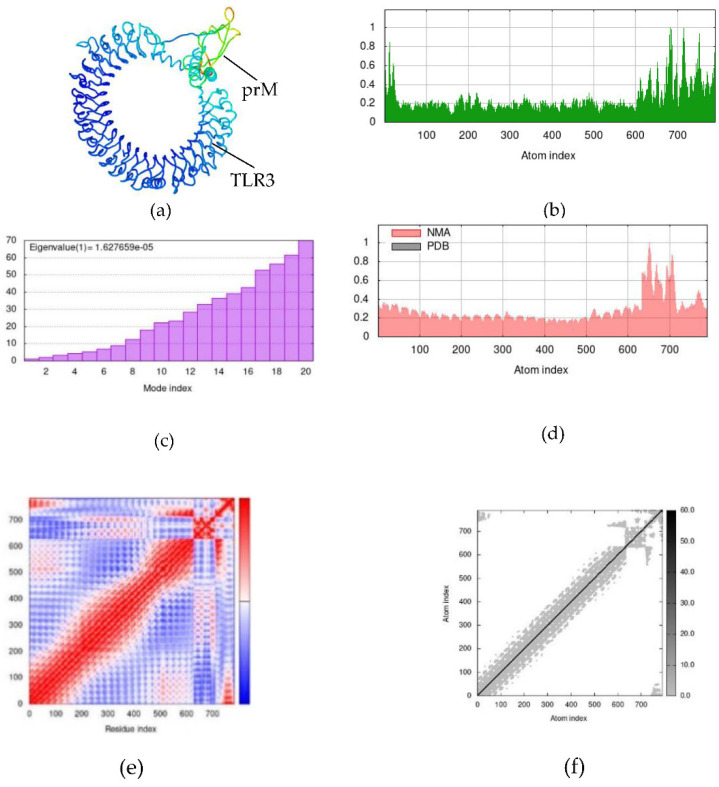
Normal mode analysis (NMA) of prM–TLR3 docking complex by iMODs. (**a**) prM and TLR3 docking complex; (**b**) main-chain deformability; (**c**) the eigenvalue; (**d**) B-factor values; (**e**) covariance map; (**f**) elastic network of model.

**Figure 9 ijms-23-13911-f009:**
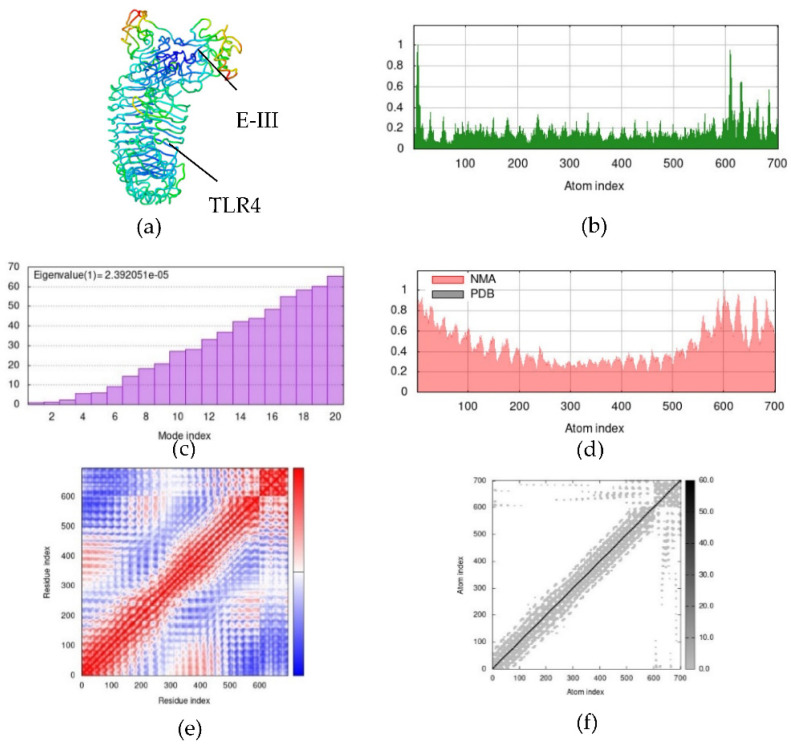
Normal mode analysis (NMA) of EIII–TLR4 docking complex by iMODs. (**a**) EIII and TLR4 docking complex; (**b**) main-chain deformability; (**c**) the eigenvalue; (**d**) B-factor values; (**e**) covariance map; (**f**) elastic network of model.

**Figure 10 ijms-23-13911-f010:**
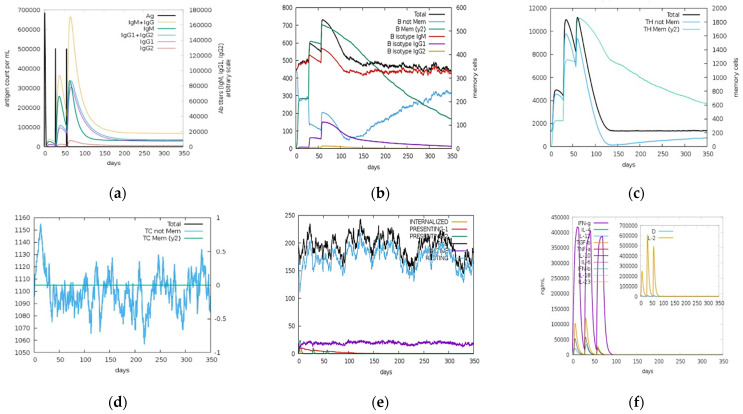
C-ImmSim server predictions for NS1 protein: (**a**) immunoglobulins (IgG1, IgG2, IgM, IgG, IgM, and antigens); (**b**) memory cell (B-cell) population; (**c**) helper T cell (HTL) population; (**d**) cytotoxic T cell (CTL) population; (**e**) macrophage count per state; (**f**) cytokine and interleukin (IL) production with Simpson index (D) of the immune response.

**Figure 11 ijms-23-13911-f011:**
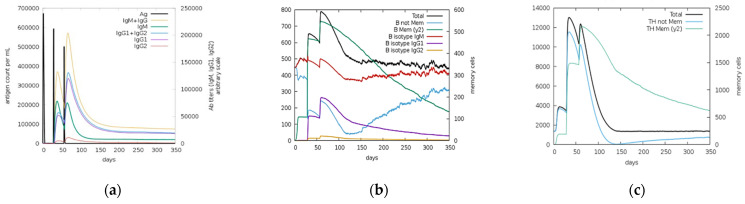
C-ImmSim server predictions for prM protein: (**a**) antigen and immunoglobulins (IgG1, IgG2, IgM, IgG, IgM, and antigens); (**b**) memory cell (B-cell) population; (**c**) helper T cell (HTL) population; (**d**) cytotoxic T cell (CTL) population; (**e**) macrophage population count per state; (**f**) cytokine and interleukin (IL) production with Simpson index (D) of the immune response.

**Figure 12 ijms-23-13911-f012:**
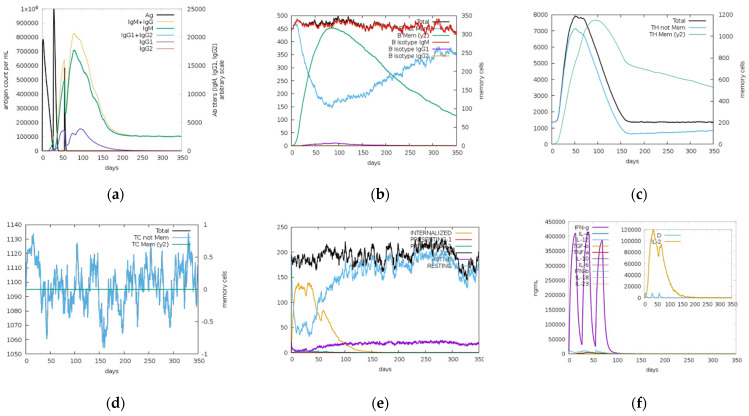
C-ImmSim server predictions for EIII protein: (**a**) immunoglobulins (IgG1, IgG2, IgM, IgG, IgM, and antigens); (**b**) memory cell (B-cell) population; (**c**) helper T cell (HTL) population; (**d**) cytotoxic T cell (CTL) population; (**e**) macrophage count per state; (**f**) cytokine and interleukin (IL) production with Simpson index (D) of the immune response.

**Figure 13 ijms-23-13911-f013:**
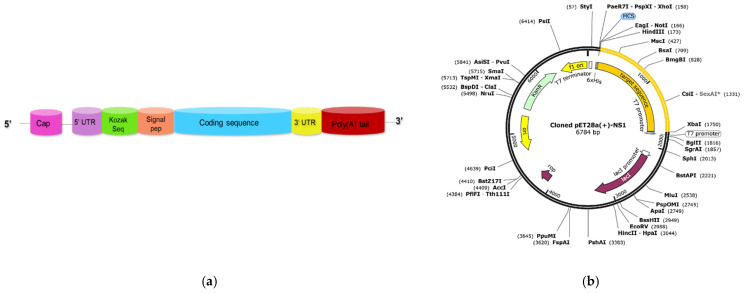
(**a**) Illustration of the complete in vitro transcribed mRNA construct with a cap, un-translated regions (UTRs), Kozak sequence, signal peptide sequence, and linked to the coding area, a poly (A) tail; (**b**) plasmid pET28a(+) containing the definitive vaccine construct (NS1–mRNA).

**Figure 14 ijms-23-13911-f014:**
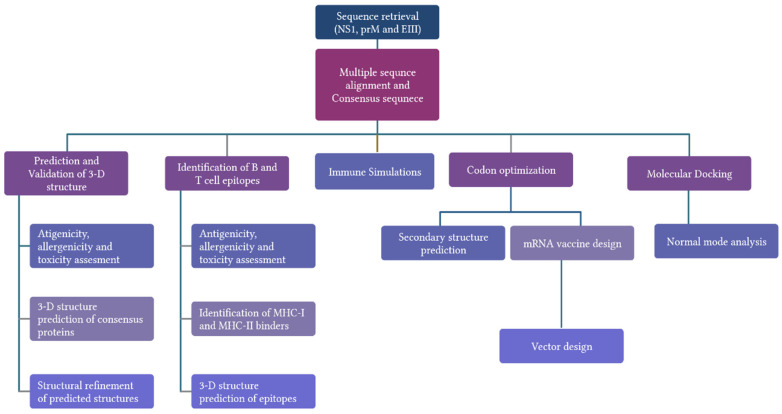
Graphical illustration of workflow of development of mRNA vaccines against dengue virus.

**Table 1 ijms-23-13911-t001:** Codon optimization of Nucleotides.

Parameters	NS1	prM	EIII
Codon adaptation index (CAI)	0.95	0.95	0.95
GC content (%)	65.0	66.2	62.8
Minimum free energy (MFE) of optimal secondary structure	−402.40 kcal/mol	−177.70 kcal/mol	−107.80 kcal/mol
Energy of thermodynamic ensemble	−416.07 kcal/mol	−182.94 kcal/mol	−112.14 kcal/mol
Free energy of centroid secondary structure	−326.03 kcal/mol	−173.70 kcal/mol	−99.50 kcal/mol

**Table 2 ijms-23-13911-t002:** Physicochemical Profiling of Linear B-Cell Epitopes.

Proteins	Epitopes	Start Position	Antigenicity	Allergenicity	Toxicity
NS1	PETAECPNTNRAWNSL	137	0.44	Negative	Negative
	YGFGVFTTNIWLKLKE	157	0.71	Negative	Negative
prM	YGTCTATGEHRREKRS	66	1.11	Negative	Negative
	AYTIGTTYFQRVLIFI	130	0.42	Negative	Negative
	GFTIMAAILAYTIGTT	121	0.66	Negative	Negative
	RVLIFILLTAVAPSMT	140	0.60	Negative	Negative
EIII	GRLITVNPIVTEKDSP	45	0.48	Negative	Negative
	NPIVTEKDSPVNIEAE	51	0.94	Negative	Negative

**Table 3 ijms-23-13911-t003:** List of selected CD8+ CTL epitopes.

Proteins	Epitopes	Alleles	Conservancy	Vaxijen Score	Allergenicity	Toxicity
NS1	ETAECPNTNR	HLA-A*68:01	100%	0.6373	Negative	Negative
	VTRLENLMWK	HLA-B*57:01	100%	0.9385	Negative	Negative
	VVSWKKKELK	HLA-A*03:01	90%	1.6108	Negative	Negative
prM	MGLETRTETW	HLA-B*58:01	100%	1.0138	Negative	Negative
	FTIMAAILAY	HLA-B3501	100%	0.4654	Negative	Negative
	DCWCNATSTW	HLA-B*58:01	100%	0.5754	Negative	Negative
EIII	EIAETQHGTI	HLA-A*02:03	100%	0.8075	Negative	Negative
	KEIAETQHGT	HLA-B*44:02	100%	0.4853	Negative	Negative

**Table 4 ijms-23-13911-t004:** List of selected CD4+ HTL epitopes.

Proteins	Epitope	Allele	Vaxijen Score	Allergenicity	Toxicity	IFN	IL4	IL10
NS1	VFTTNIWLKLKERQD	HLA-DRB3*02:02	0.7717	−ve	−ve	+ve	+ve	+ve
prM	RHPGFTIMAAILAYT	HLA-DRB1*04:01	0.4237	−ve	−ve	+ve	+ve	+ve
EIII	KIPFEIMDLEKRHVL	HLA-DRB5*01:01	1.1746	−ve	−ve	−ve	+ve	+ve

**Table 5 ijms-23-13911-t005:** Docking of selected CTL epitopes with their corresponding MHC-I alleles.

Epitopes	Alleles	Protein Structure Similarity Score	Interaction Similarity Score	Estimated Accuracy
ETAECPNTNR	HLA-A*68:01	0.993	138.0	0.928
VTRLENLMWK	HLA-B*57:01	0.996	297.0	1.000
VVSWKKKELK	HLA-A*03:01	0.956	193.0	1.000
MGLETRTETW	HLA-B*58:01	0.991	252.0	1.000
FTIMAAILAY	HLA-B3501	0.991	230.0	1.000
DCWCNATSTW	HLA-B*58:01	0.964	227.0	1.000
EIAETQHGTI	HLA-A*02:03	0.985	184.0	1.000
KEIAETQHGT	HLA-B*44:02	0.985	147.0	0.940

**Table 6 ijms-23-13911-t006:** Docking of selected HTL epitopes with their corresponding MHC-II alleles.

Epitopes	Alleles	Members	Representative	Weighted Score
**VFTTNIWLKLKERQD**	HLA-DRB3*02:02	105	Center	−774.4
		105	Lowest Energy	−915.1
**RHPGFTIMAAILAYT**	HLA-DRB1*04:01	479	Center	−978.8
		479	Lowest Energy	−988.4
**KIPFEIMDLEKRHVL**	HLA-DRB5*01:01	252	Center	−898.4
		252	Lowest Energy	−1101.2

**Table 7 ijms-23-13911-t007:** Docking between immune receptors associated with dengue virus, i.e., DC-SIGN, mannose-binding receptor (MR), TLR3, and TLR4, and candidate vaccine proteins.

Ligand	Receptor	Members	Representative	Weighted Score
**NS1**	DC-SIGN	217	Lowest Energy	−1358.8 kcal/mol
	MR	110	Lowest Energy	−991.9 kcal/mol
	TLR3	14	Lowest Energy	−1120.1 kcal/mol
	TLR4	15	Lowest Energy	−1013.9 kcal/mol
**prM**	DC-SIGN	110	Lowest Energy	−1164 kcal/mol
	MR	110	Lowest Energy	−1017.2 kcal/mol
	TLR3	65	Lowest Energy	−1149.8 kcal/mol
	TLR4	38	Lowest Energy	−1257.7 kcal/mol
**EIII**	DC-SIGN	100	Lowest Energy	−862.2 kcal/mol
	MR	78	Lowest Energy	−753.3 kcal/mol
	TLR3	118	Lowest Energy	−971.9 kcal/mol
	TLR4	103	Lowest Energy	−892.3 kcal/mol

## Data Availability

Not applicable.

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
