# Peer review of "Engineering Modified mRNA-Based Vaccine against Dengue Virus Using Computational and Reverse Vaccinology Approaches"

_ijms, 2022, doi:10.3390/ijms232213911_

Round 1
Reviewer 1 Report
This manuscript by Mukhtar et al. examines the NS1, prM and EIIIof four serotypes of Dengue virus from Pakistani iso-lates for developing mRNA vaccine using computational approaches. The anthers perform the codon optimization of conserved sequence and identify the B-cell and T-cell epitopes. They also design the vaccine construct as the vaccinal candidate.
The authors have performed an amount of work and my impression is that they have tried very hard to present the results clearly, but at places this breaks down due to poor description of the results. Instead of writing an interesting story, the authors listed too many software and promoters in the result section. To strengthen the findings of the paper, the reviewer has several concerns that will need to be addressed.
Major comment
1. In the title, the authors claimed that they used the machine learning to engineer mRNA vaccine. However, the authors only mentioned the machine learning in the in silico immune simulation part without any details. They failed to explain how important of machine learning to this manuscript.
2. To show the B-cell epitopes directly, I suggest the authors label the epitopes in the 3D structure of NS1, prM and EIII.
3. In Fig. 5, can authors label the proteins in the docking complex and use same color for the same protein, for example, using the color for all the NS1 protein.
4. In Fig. 6a, 7a, 8a, label the proteins in the docking complex.
5. In Fig. 9, 10 and 11, the legends are overlapped within the curve. I suggest the authors remake the figures.
6. The data in Fig. S6 is too small to read. Please enlarge the numbers and labels in the figure.
Author Response
Major comment
1. In the title, the authors claimed that they used the machine learning to engineer mRNA vaccine. However, the authors only mentioned the machine learning in the in silico immune simulation part without any details. They failed to explain how important of machine learning to this manuscript.
Response: Thank you for appreciating our work and rewarding suggestions. As we have used multiple Machine Learning (ML) tools and approaches for designing the vaccine against dengue virus by structural modeling of target consensus protein, molecular docking and Normal mode analyses of the modeled protein structure, selection of B and T cell epitopes and their docking simulations study and Immune simulations for vaccine development. We have also modified the title of the article in the revised manuscript by adding the word “Computational” instead of “Machine learning” to avoid any conflict.
- To show the B-cell epitopes directly, I suggest the authors label the epitopes in the 3D structure of NS1, prM and EIII.
Response: The said suggestion has been incorporated and the candidate proteins have been color marked to show the specific epitopes on the 3-D structure of NS1, prM and EIII proteins.
- In Fig. 5, can authors label the proteins in the docking complex and use same color for the same protein, for example, using the color for all the NS1 protein.
Response: We have modified the figure by labelling the proteins in the docking complex and using same color for the same protein.
- In Fig. 6a, 7a, 8a, label the proteins in the docking complex.
Response: We have modified the figures as per reviewer’s suggestion.
- In Fig. 9, 10 and 11, the legends are overlapped within the curve. I suggest the authors remake the figures.
Response: We absolutely agree with the reviewer, but unfortunately the figures are system-generated (by C-ImmSim server) and cannot be recreated. However, we have improved the quality and increased the size of figures for clarity.
- The data in Fig. S6 is too small to read. Please enlarge the numbers and labels in the figure.
Response: We have enlarged the numbers and labels in the figures as suggested by the reviewer.
Reviewer 2 Report
The authors present work titled “Engineering modified mRNA-based vaccine against Dengue virus using machine learning and reverse vaccinology”. This work appears to be purely computational in nature with several tools, web-servers, and machine learning approaches. The results obtained from their work forms a good basis to be further validated using experiments. Overall the topic is interesting, however there are some concerns that need to be addressed before the paper can be accepted.
1. It will be useful to provide a flow-chart / diagram of procedure followed to propose mRNA-based vaccine against Dengue virus.
2. The authors performed two docking simulations, one between the consensus protein and Dengue receptors and the other between the selected epitopes and their respective MHC alleles. However, binding energy information obtained from docking experiments is not provided. It is therefore not clear how can they say if an epitope is good/moderate/poor binder.
3. No molecular dynamics (MD) simulations has been performed by the authors. They have only done normal mode analysis using the iMods server, which does not perform MD simulations. So the authors should correct this misleading information.
4. It would have been nice if the authors could have performed MD simulations for at least one of the systems to check if the complex predicted by docking is stable.
5. The authors used C-ImmSim server to predict and estimate the real-life immunogenic profile of the mRNA vaccine. It would be useful to provide the confidence score of the method used in the server. How close can the server predict to an experimental data?
Author Response
- It will be useful to provide a flow-chart / diagram of procedure followed to propose mRNA-based vaccine against Dengue virus.
Response: We have incorporated a graphical illustration of workflow in the revised manuscript for better understanding of the procedures followed to propose mRNA based vaccines against Dengue Virus.
- The authors performed two docking simulations, one between the consensus protein and Dengue receptors and the other between the selected epitopes and their respective MHC alleles. However, binding energy information obtained from docking experiments is not provided. It is therefore not clear how can they say if an epitope is good/moderate/poor binder.
Response: We agree with the reviewer and have incorporated the data obtained from the docking simulations in tabulated form. The lowest energy and other parameters for good/moderate/poor binders have been mentioned in the selection of final epitopes.
- No molecular dynamics (MD) simulations has been performed by the authors. They have only done normal mode analysis using the iMods server, which does not perform MD simulations. So the authors should correct this misleading information.
Response: We absolutely agree with the reviewer and the corrections have been made. The title “MD simulations” have now been changed to “Normal Mode Analysis (NMA)” in the revised manuscript.
- It would have been nice if the authors could have performed MD simulations for at least one of the systems to check if the complex predicted by docking is stable.
Response: We totally agree with the reviewer’s suggestion. Since the docking complexes are massive and performing MD simulations on them is a time-taking process. We are also working on wet-lab validations of the proposed vaccine, hence we have intended to publish the results of MD simulations in the subsequent manuscript along-with wet-lab validation results.
- The authors used C-ImmSim server to predict and estimate the real-life immunogenic profile of the mRNA vaccine. It would be useful to provide the confidence score of the method used in the server. How close can the server predict to an experimental data?
Response: We agree with the reviewer’s comment and have keenly looked for the validation of C-ImmSim server. There are a huge number of paper reporting the in silico estimation of immunogenic response generated by the vaccine, nevertheless there is no confidence score available from the wet-lab validation of the predicted results. Since we are planning on wet-lab validation of our predicted results as well, we will keep this point in mind while compiling the data in following publication.
Reviewer 3 Report
The authors reported the design and computational characterization of mRNA vaccine against dengue virus. It is impressive to use all different kinds of computational tools to design and analyze the sequence, although many of them is based on web server. Considering no wet experiment to prove the design, I strongly suggest the authors should add positive control(experimentally/clinically used mRNA vaccine) to as many computational analysis as possible, including (1) mRNA structure and minimum free energy (2) Docking with respect to binding free energy or other quantitative parameter (3) MD simulation (4) Immune simulation. It will be more convincing to include the positive control together with your new design and discuss their difference if there is. Also, audience will develop a better sense of how to relate the prediction with the physical data.
Minors:
Could you use alphaFold or MegaFold for structure prediction since they has been reported to be more accurate than trRosetta.
Figure 6 figure number is merged to figure.
Please add “computational” to the title since there are no wet experiment data.
Author Response
Minors Comments:
1. Could you use alphaFold or MegaFold for structure prediction since they has been reported to be more accurate than trRosetta.
Response: We strongly agree with the reviewer’s comment. In this study we have used TrRosetta for the predictions of 3-D structures and Galaxy Refine for the refinements of selected models. The refined models were then checked for their Errat score and Ramachandran plots which suggested that the predicted models are the best possible structures. We have also considered the reviewer’s suggestion and performed the structural predictions using “alphaFold” and subjected the predicted models to saves server for their Errat score and Ramachandran plots. The results then generated by the Saves server were compared with the previously mentioned results from TrRosetta. The comparison suggested that the structures predicted by TrRosetta and refined by Galaxy refine were more accurate in terms of Errat score and Ramachnadran plots.
- Figure 6 figure number is merged to figure
Response: We have modified the Figure according to the reviewer’s suggestion.
- Please add “computational” to the title since there are no wet experiment data.
Response: We have included the word “Computational” in the title of the revised manuscript.
Round 2
Reviewer 1 Report
The requested changes to the figures are sufficient. As a result, the quality of the paper is substantially improved.
Author Response
Response: Thank you very much for appreciating our work.
Reviewer 2 Report
The authors have revised the paper addressing most of my comments.
Comment 1: In the abstract, please remove MD simulations.
Author Response
Response: Thank you very much. The word “MD simulations” have been removed from abstract and key words.
Reviewer 3 Report
All comments has been addressed.
Author Response
Response: Thank you very much.